# Circulating Adipokines and Hepatokines Serve as Diagnostic Markers during Obesity Therapy

**DOI:** 10.3390/ijms232214020

**Published:** 2022-11-14

**Authors:** Andreas Schmid, Miriam Arians, Monika Burg-Roderfeld, Thomas Karrasch, Andreas Schäffler, Martin Roderfeld, Elke Roeb

**Affiliations:** 1Department of Internal Medicine III, Giessen University Hospital, Klinikstr. 33, 35392 Giessen, Germany; 2Department of Gastroenterology, Justus Liebig University, Klinikstr. 33, 35392 Giessen, Germany; 3Hochschulen Fresenius GmbH, University of Applied Sciences, 65510 Idstein, Germany

**Keywords:** obesity, Roux-en-Y gastric bypass (RYGB), low-calorie diet (LCD), adipokine, hepatokine, ANGPTL4, CCL2, GDF15, GPNMB, IGFBP6

## Abstract

Allocation of morbidly obese patients to either conservative therapy options—such as lifestyle intervention and/or low-calorie diet (LCD)—or to bariatric surgery—preferably sleeve gastrectomy or Roux-en-Y gastric bypass (RYGB)—represents a crucial decision in order to obtain sustainable metabolic improvement and weight loss. The present study encompasses 160 severely obese patients, 81 of whom participated in an LCD program, whereas 79 underwent RYGB surgery. The post-interventional dynamics of physiologically relevant adipokines and hepatokines (ANGPTL4, CCL5, GDF15, GPNMB, IGFBP6), as well as their correlation with fat mass reduction and improvement of liver fibrosis, were analyzed. Systemic GDF15 was characterized as an excellent predictive marker for hepatic fibrosis as well as type 2 diabetes mellitus. Of note, baseline GDF15 serum concentrations were positively correlated with NFS and HbA1c levels after correction for BMI, suggesting GDF15 as a BMI-independent marker of hepatic fibrosis and T2D in obese individuals. Specific GDF15 cut-off values for both diseases were calculated. Overall, the present data demonstrate that circulating levels of specific adipokines and hepatokines are regulated with therapy-induced fat loss and metabolic improvement and might, therefore, serve as biomarkers for the success of obesity therapy strategies.

## 1. Introduction

The metabolic syndrome—comprising obesity and associated physiological dysfunctions, such as type 2 diabetes mellitus (T2D), hypertension, and dyslipidemia—represents a severe and world-wide public health problem in children, adolescents, and adults [1,2] with an epidemic status [3]. Most importantly, immune-metabolic mechanisms in visceral obesity—including adipocyte hypertrophy, a pro-inflammatory shift in secretory protein profile, and pro-inflammatory macrophage polarization—summarized as metaflammation [4], favor the onset of T2D and associated metabolic disorders. Furthermore, obesity-induced dyslipidemia and dysregulation of hepatic lipid and carbohydrate metabolism can severely impair liver integrity and result in liver fibrosis [5]. Prevalence of non-alcoholic fatty liver disease (NAFLD)—including advanced non-alcoholic steatohepatitis (NASH) as well as hepatic fibrosis and cirrhosis—is strongly associated with obesity [6]. An updated clinical S2k guideline on NAFLD diagnosis and treatment was published recently [7].

Various strategies have been established during recent decades to induce a sustainable reduction of excessive body fat, as well as to restore dysregulated metabolic functions in morbid obesity [8,9]. Conservative approaches most commonly combine strategies of dietary and lifestyle intervention. Among these, low-calorie diets (LCD) applying a strong caloric restriction have proven effective both in terms of metabolic improvement and body weight loss [10,11]. Compared to conservative therapy options, bariatric surgery appeared to be more effective and sustainable in therapeutic fat and weight loss [12,13]. Among different surgical strategies for the treatment of morbid obesity, the most commonly applied is the Roux-en-Y gastric bypass (RYGB) [14] which combines the weight-reducing mechanisms of nutritional restriction and malabsorption.

Out of endocrine proteins secreted by adipose tissue—referred to as adipokines—angiopoietin-like (ANGPTL) 4 has an important role in the regulation of lipid metabolism. Originally described as fasting-induced adipose factor (FIAF), mainly expressed in adipose tissue and liver [15], ANGPTL4 expression is increased in the fasting state [16]. Representing an inhibitor of lipoprotein lipase (LPL) activity, the regulation of lipid homeostasis is considered a major physiological function of ANGPTL4 [17].

C-C chemokine motif ligand (CCL) 5—also referred to as RANTES—is a chemokine with chemotactic abilities involved in various inflammatory pathologies via leukocyte recruitment and macrophage polarization [18,19]. Whilst being associated with hepatic steatosis [20], the regulation and role of CCL5 in obesity requires further elucidation. Its well-characterized pro-inflammatory properties make it a putative factor in obesity-related metaflammation, with a potential role in monocyte recruitment to adipose tissue.

Growth differentiation factor-15 (GDF15) represents a hormone within the transforming growth factor-β (TGF-β) superfamily and a marker of cellular stress. While GDF15 expression was detected at low levels in numerous tissues, it was found to be strongly increased in activated macrophages [21]. Of note, a high-fat diet (HFD) in mice resulted in elevated GDF15 expression in adipose tissue [22], implying a role in signaling metabolic stress. A recent study identified the association of increased GDF15 expression with the severity of nonalcoholic fatty liver disease (NAFLD) by transcriptomic profiling [23]. Since the expression of its receptor GFRAL is limited to the hindbrain, GDF15 is able to function as a stress signal for metabolic dysregulation and disease to the central nervous system [21]. Mediated by the receptor heterodimer GFRAL-RET, GDF15 regulates feeding behavior by the reduction of appetite [24].

The type-I transmembrane protein glycoprotein non-metastatic melanoma protein B (GPNMB)—also referred to as osteoactivin—is released in its soluble form upon protease cleavage [25]. It is expressed in various cell-types, including macrophages and dendritic cells, and is involved in cancer and inflammatory pathologies, including steatohepatitis and colitis [25,26], exhibiting a rather anti-inflammatory character in most contexts. Importantly, GPNMB is upregulated in obesity-related NAFLD, exerting beneficial impact by reducing oxidative stress [27]. GPNMB was also reported to protect against obesity-related inflammation by inhibition of inflammatory cytokine secretion from macrophages [28].

Hepatic expression of insulin-like growth-factor binding protein (IGFBP)-6—a transport protein for insulin-like growth-factor 2 (IGF2)—was recently reported to be positively associated with steatosis and fibrosis in NAFLD [29].

Despite the constantly growing quantity of data on metabolic syndrome, obesity, and accompanying liver diseases, knowledge of the precise role of adipokines and cytokines with regulatory metabolic functions in the crosstalk of adipose tissue, liver, and the regulation of dietary behavior in therapy-induced weight loss remains unclear so far.

In front of this background, and in order to further clarify this issue, our present study investigated data from a large and well-characterized obesity cohort comprising both diet-based and bariatric strategies for sustained weight reduction. In particular, it aimed to investigate 

-basal circulating concentrations of ANGPTL4, CCL5, GDF15, GPNMB, and IGFBP6,-the dynamic changes of these systemic levels during weight loss, either during LCD or following RYGB surgery, and-the correlation of these changes with therapy-induced body fat loss, improvement of T2D, and reduced risk of liver fibrosis.

## 2. Results

### 2.1. Study Characteristics

General anthropometric, biochemical, and pathophysiological characteristics of the study cohorts have been reported and extensively discussed in a recent publication [30]. A brief summary of the most important parameters at the study base-line and a 12 month follow-up is provided in Table 1, as published previously [30]. Briefly, considerable weight loss was achieved by both therapeutical strategies, LCD and RYGB (Table 1).

### 2.2. Dynamics of Systemic Cytokine Concentrations during Weight Loss

Circulating concentrations of ANGPTL4, CCL5, GDF15, GPNMB, and IGFBP6 were quantified via ELISA at study-baseline and after 3, 6, and 12 months of RYGB surgery (n = 79) or the start of LCD program (n = 81), respectively (Figure 1). Basal ANGPTL4 levels were 859 ± 1403 ng/mL within the LCD cohort and 667 ± 862 ng/mL among bariatric surgery cohorts. Neither RYGB surgery nor LCD significantly affected mean ANGPTL4 concentrations during the 12 months of weight loss (Figure 1A). CCL5 was significantly reduced by the conservative therapy as well as by the surgical intervention (34.19 ± 20.24 vs. 7.13 ± 11.76 ng/mL and 39.36 ± 26.80 vs. 7.72 ± 11.96 ng/mL, respectively) (Figure 1B). Similarly, circulating levels of GDF15 and IGFBP6 were decreased following both LCD and RYGB (GDF15: 378 ± 188 vs. 295 ± 195 pg/mL and 481 ± 416 vs. 319 ± 200 pg/mL, respectively; IGFBP6: 347 ± 95 vs. 280 ± 169 ng/mL and 362 ± 94 vs. 295 ± 181 ng/mL, respectively) (Figure 1C,D). Unlike these findings, systemic GPNMB were increased after RYGB (24.9 ± 30.7 vs. 55.9 ± 109.4) and LCD during 12 months after base-line (23.9 ± 23.2 vs. 36.0 ± 77.6 ng/mL) without statistical significance (Figure 1E). ROC curve analysis was performed for the 12 month changes of all cytokines regarding the classification of individuals with high (upper 50%) or low body fat loss (lower 50%) that was applied to the whole study cohort (LCD + RYGB) [30]. The results indicate that none of the analyzed proteins represent an applicable classifier for the discrimination of patients with higher or lower body fat loss (Figure 1F).

### 2.3. Relation of Weight Loss Associated GDF15 Dynamics to Liver Fibrosis

Overall, circulating GDF15 concentrations were increased in 40 patients and decreased in 120 patients within 12 months after RYGB surgery and after the beginning of LCD, respectively (*p* < 0.001 each) (Figure 2A). Individuals with increasing GDF15 levels during weight loss therapy started with significantly lower basal concentrations than those offering GDF15 decline (295 vs. 474 pg/mL; *p* < 0.001). After 12 months, this difference in serum GDF15 was reversed (459 vs. 256 pg/mL; *p* < 0.001) by a significant raise and decline, respectively (*p* < 0.001 each) (Figure 2B). Of note, in the 12 month follow-ups, the mean NAFLD fibrosis score (NFS) was higher in patients with decreased serum GDF15 (−1.85 vs. −2.32; *p* = 0.028) (Figure 2C). Further, increasing GDF15 levels were accompanied by increasing FIB-4 scores (*p* < 0.001), resulting in a higher mean FIB-4 compared to patients with decreased GDF15 after 12 months (*p* = 0.006) (Figure 2D).

From available clinical data assessed at the study base-line, 64 patients could be characterized as non-fibrotic (NFS < −1.445, 46 LCD, and 18 RYGB patients) and 14 as suffering from advanced fibrosis (NFS > 0.675, 8 LCD, and 6 RYGB patients) (Figure 3A). Overall, the subjects with initial advanced fibrosis exhibited considerably higher basal GDF15 serum levels (688 vs. 339 pg/mL; *p* < 0.001) and also experienced a transient GDF15 increase during the first 3 months of weight loss therapy (Figure 3B). The differences in base-line GDF15 concentrations and the dynamics during the following 12 months were observed for LCD as well as RYGB patients (Figure 3C,D). Mean NFS values, being considerably higher in subjects with fibrosis (*p* < 0.001), were not significantly affected by weight loss (Figure 3E). ROC curve analysis confirmed that GDF15 represents an excellent marker for the discrimination of individuals with advanced hepatic fibrosis (NFS > 0.76; n = 14) and without hepatic fibrosis (NFS < −1.455; n = 64) at study base-line (AUC = 0.910) (Figure 3F). A GDF15 cut-off value of 415.41 pg/mL predicted advanced fibrosis with 92.9% sensitivity and 85.9% specificity. This strongly predictive character for hepatic fibrosis was exclusively observed for GDF15 but not for ANGPTL4, CCL5, GPNMB, and IGFBP6.

Partial correlation analysis revealed a positive, BMI-independent correlation of base-line GDF15 serum levels with NFS values (r = 0.392, *p* < 0.001).

### 2.4. GDF15 and Hypertension

At the study base-line, a total of 91 patients with hypertension exhibited elevated GDF15 serum concentrations (*p* < 0.001) compared to those without hypertension (n = 66) (392 vs. 165 pg/mL; *p* < 0.001) (Figure 4A). Dividing the study cohort into subgroups defined by BMI, we found that this difference was significant for patients with a BMI higher than 45 kg/m^2^ (n = 105) but not for individuals with a lower extent of obesity (BMI ≤ 45; n = 55). At the 12 month follow-up, the number of patients with hypertension was 56 and there was no longer a significant difference in GDF15 levels when compared to subjects without hypertension (Figure 4B). GDF15 dynamics during weight loss—with a non-significant trend to slightly elevated concentrations after 3 months and significant reduction after 12 months—were independent of hypertension and therapeutical strategy (Figure 4C–E). ROC curve analysis of both study sub-cohorts indicated that serum GDF15 is a moderately applicable marker for patients’ classification regarding hypertension (LCD: AUC = 0.716; RYGB: AUC = 0.689) (Figure 4F).

### 2.5. GDF15 as a Serum Marker of Type 2 Diabetes Mellitus

Circulating concentrations of GDF15 were positively correlated with HbA1c levels at the study base-line in both sub-cohorts (LCD: rho = 0.284, *p* = 0.010; RYGB: rho = 0.523, *p* < 0.001). This correlation was independent of BMI in the RYGB group as was confirmed by partial correlation analysis (r = 0.556, *p* < 0.001) (LCD + RYGB: r = 0.432, *p* < 0.001). Changes in serum HbA1c until the 12 month follow-up were positively correlated with GDF15 changes among RYGB patients (rho = 0.270, *p* = 0.028) but not within the LCD group. Furthermore, base-line GDF15 serum concentrations were negatively correlated with changes in HbA1c levels (rho = −0.346, *p* < 0.001) at the 12 month follow-up, unlike the other cytokines investigated in the present study.

ROC curve analysis indicated GDF15 as an acceptable marker for T2D classification within the whole study cohort at base-line (AUC = 0.780) (Figure 5A). Importantly, GDF15 proved to be a significantly better T2D classifier in the surgical than in the LCD sub-cohort (AUC: 0.875 vs. 0.637; *p* = 0.009) (Figure 5B). For the RYGB group, a GDF15 cut-off value of 370 pg/mL might be suggested, resulting in a sensitivity of 91.3% and a specificity of 71.4% for T2D classification.

## 3. Discussion

The present study comprises data for a panel of physiologically relevant adipokines and hepatokines in a large cohort of obese patients undergoing either low-caloric dietary or surgical (RYGB) therapy. We investigated systemic base-line concentrations as well as dynamics after the beginning of the particular intervention.

Overall, systemic CCL5, GDF15, and IGFBP6 levels declined following 12 months of anti-obesity therapy and weight loss, irrespective of a conservative or bariatric approach. Serum GPNMB concentrations, however, were not significantly affected. ANGPTL4 exhibited a non-significant trend of increasing levels upon RYGB surgery.

The observed reduction of CCL5 concentrations were presumably due to a general improvement of the obesity-related, low-grade inflammation state [4] during and as a consequence of considerable weight loss. Similarly, decreasing IGFBP6 levels might reflect the beneficial metabolic changes accompanying weight loss and the reduction of fat mass. Overall, the observed dynamics in systemic cytokine levels did not significantly predict the extent of body fat loss.

The negative regulation of circulating GDF15 during weight loss therapy is of particular interest, due to its regulatory role in nutritional behavior and potential implications for obesity therapy [31], as well as its association with NAFLD [23]. It appears reasonable that considerable loss of weight and fat mass is accompanied by a significant reduction of elevated, obesity-related GDF15 levels. Since GDF15 inhibits food intake by GFRAL-mediated signal transduction [32], its decline might be induced by regulatory mechanisms reacting to the strong and prolonged weight loss, thus displaying a diminished necessity of appetite suppression. Of note, no difference regarding GDF15 dynamics was observed between the LCD cohort receiving conservative dietary intervention and the patients undergoing RYGB surgery. The latter were facing a dramatic, quantitative restriction of food intake following surgical intervention which, however, apparently did not additionally affect appetite regulating GDF15. Thus, the observed decrease of circulating GDF15 concentrations is predominantly caused by the weight loss per se and independent of the therapeutic strategy applied.

Since morbid obesity is frequently associated with severe liver disease [6], the present study focused on a putative association of weight loss-related systemic cytokine dynamics with scores of liver integrity and fibrosis—NAFLD fibrosis score (NFS) and the fibrosis index FIB-4—being calculated from basal clinical data. Overall, a moderate decline of NFS values was observed during weight loss within 12 months independently of GDF15 decrease or increase, respectively. Meanwhile, there was a raise in the mean FIB-4 score which was clearly associated with increasing GDF15 levels. Taken together, these data imply that decreasing rather than increasing systemic GDF15 concentrations might indicate amelioration of liver integrity after 12 months of weight loss. Besides this moderate correlation, advanced liver fibrosis—diagnosed via NFS—was observed to be associated with considerably elevated basal GDF15 concentrations. This finding is consistent with the outcome of previous studies [23,33] and confirms the predictive value of GDF15 as a clinical marker for hepatic fibrosis. Of note, base-line GDF15 serum concentrations positively correlated with NFS after correction for BMI, suggesting GDF15 as a BMI-independent marker of hepatic fibrosis in obese individuals. The observed GDF15 dynamics during the phase of significant weight loss—including a transient increase during the initial three months—did not significantly differ between conservative and surgical therapy. Based on our results from ROC curve analysis, a serum GDF15 concentration of 415.41 pg/mL might be suggested as a cut-off value for discrimination of advanced hepatic fibrosis versus the absence of fibrosis. Elevated GDF15 levels in hepatic fibrosis strongly argue for a significant role of this cytokine in the pathogenesis of the disease. As reported previously by Kim et al., GDF15 attenuates non-alcoholic steatohepatitis by inhibition of pro-fibrotic gene expression in mice [34]. As a regulator of appetite and nutritional behavior, elevated systemic GDF15 concentrations might further represent a mechanism for amelioration of hepatic fibrosis by restricted food intake and, therefore, a reduced fat load in the liver. Of particular importance, data supporting the putative role of GDF15 as an obesity-independent marker and predictor of hepatic fibrosis remain scarce so far.

The other cytokines investigated in the present study did not exhibit any significant predictive character regarding liver fibrosis. As a consequence, we further focused on GDF15 and its correlation with physiological and metabolic aspects.

Patients with arterial hypertension were characterized by significantly elevated basal serum concentrations of GDF15, which is in line with its property as a biomarker of epithelial stress and cardiovascular risk in previous studies [35,36]. A recent study proposed a GDF15 cut-off value for the diagnosis of coronary artery disease [37]. Importantly, our present data indicate that the observed elevation of GDF15 levels in hypertension depends on BMI and might be an attribute of excessively obese individuals (BMI > 45 kg/m^2^). During weight loss, individuals with initial hypertension experienced an overall stronger decline of GDF15 levels despite largely congruent dynamics during the observed time span of 12 months. As a result, there was no significant difference in GDF15 at the end of this period in our study. Thus, the decline in GDF15 levels might reflect the generally improved cardiovascular situation associated with the significant reduction of body weight and fat mass being achieved in both therapeutical approaches. Overall, our present data confirm the potential of GDF15 as a reliable biomarker for hypertension in severe obesity.

At study base-line, GDF15 serum concentrations positively correlated with HbA1c levels—a marker for long-term elevated systemic glucose levels and insulin resistance. This correlation was independent of BMI, as we confirmed by partial correlation analysis. Of note, basal GDF15 levels were also negatively correlated with HbA1c changes during 12 months after the beginning of intervention. This finding suggests circulating GDF15 as a predictor of weight loss associated T2D amelioration. Furthermore, ROC curve analysis indicated that GDF15 represents an appropriate marker for T2D. Most interestingly, this was particularly the case for the sub-cohort of subjects designated to receive bariatric surgery. Since these patients exhibited a significantly higher mean BMI than those participating in the conservative dietary program [30], this finding indicates that the correlation of GDF15 with T2D might be BMI-dependent and predominantly exists in individuals with advanced obesity. Further studies comparing large cohorts of subjects from a broad range of BMI will be necessary in order to address this issue. Elevated GDF15 concentrations in T2D patients might be considered as an anti-diabetic feed-back mechanism due to restricting effects on food intake and subsequent beneficial impact on weight loss [38] and insulin resistance. Of note, we observed a positive correlation in post-surgical serum GDF15 dynamics with changes in HbA1c levels whilst such a correlation was absent in the LCD sub-cohort. This observation suggests that weight loss associated dynamics in systemic GDF15 and HbA1c might specifically occur following RYGB but not during dietary intervention.

To the best of our knowledge, the present study is the first to investigate systemic GDF15 concentrations in a large cohort of obese patients receiving either dietary or bariatric surgery, during a time span of 12 months. The study design allows the identification of potential correlations of conservative and surgical obesity therapies with physiological outcome parameters. We were able to identify correlations of GDF15 concentrations with glucose metabolism, liver integrity, and cardiovascular health. Importantly, cut-off values for the classification of patients regarding T2D and hepatic fibrosis were recommended, thus providing potential diagnostic tools for the clinical characterization of obese individuals.

## 4. Materials and Methods

### 4.1. ROBS (Research in Obesity and Bariatric Surgery) Study Cohort

Serum samples were collected from the *ROBS* (*Research in Obesity and Bariatric Surger*y) study cohort [39]. *ROBS* is an open-label, non-randomized, monocentric, prospective, and observational (explorative and confirmatory) study of patients routinely undergoing either bariatric surgery (gastric sleeve or Roux-en-Y gastric bypass) or a low-calorie formula diet (LCD) in the tertiary care centre at the University Hospital Giessen, Germany. The detailed information about this study cohort can be drawn from a previous publication [39]. The present study comprises data for *ROBS* subjects who completed visits V (base-line), V3, V6, and V12 (3, 6, and 12 month follow-ups) and, thus, represents an extension of the study cohort introduced by Brock et al. in 2019 [39].

The study was approved by the local ethical committee at the University of Giessen, Germany (file: *AZ 101/14*). All patients gave their informed consent and were informed about the aim of the study. Data anonymization and privacy policy were accurately applied.

In the present study, a total of 160 obese patients were enrolled and received either a RYGB surgery (n = 79) or underwent an LCD program as a conservative therapy approach (n = 81). Liver fibrosis scores (NFS, FIB-4) were assessed according to current guidelines and applying established calculation formula [40,41,42] as was reported recently [30]. Detailed information on this sub-cohort of the *ROBS* study has been published previously [30].

### 4.2. Data Collection

Data collection was performed at four different time points, before RYGB surgery or at the beginning of dietary intervention (V0) and after 3, 6, and 12 months (V3, V6, and V12). The examination of the patients included an anthropometric assessment, collection of clinical and psychological data as well as medication, smoking habits, and nutritional status, routine laboratory examination (CRLE), and protein quantification in blood serum samples. The individual parameters for the study cohort have been published previously [30].

12 month follow-up visits were chosen for the analysis of physiological parameters due to the observed considerable decrease in body weight and body fat percentage in both study cohorts during this time span [30], which was hypothesized to be associated with significant beneficial effects on liver fibrosis, T2D, and hypertension. Established cut-off values of the NAFLD fibrosis score (NFS) were applied in order to discriminate patients with advanced (NFS > 0.675) and without liver fibrosis (NFS < −1.455) [43]. Patients with T2D were defined by previous medical diagnosis or by cut-off values for either HbA1c levels (>6.5%), fasting glucose (>110 mg/dL), or impaired glucose tolerance (>180 mg/dL glucose after 2 h in standardized oral glucose tolerance test). Individuals with arterial hypertension were identified by medical diagnosis and antihypertensive drug medication.

### 4.3. Quantification of Systemic Adipokine and Hepatokine Levels

Circulating GDF15 was analyzed in blood serum as a currently discussed promising factor which might serve as a therapeutic target for obesity [38] and GPNMB because of its important features in the crosstalk between liver and adipose tissue [44]. The regulation of IGFBP6, ANGPTL4, and CCL5 during weight loss was described before [45], but their predictive potential for liver fibrosis or T2D has not been addressed yet. Blood samples of all study subjects were collected at base-line as well as 3, 6, and 12 months after the start of therapeutic intervention (LCD program or RYGB surgery, respectively). In the RYGB subgroup, additional blood sampling was done 3–4 days post-surgery. Blood serum was prepared by centrifugation (4000 rpm, 15 min, 4 °C) and circulating protein concentrations (Angptl4, CCL2, GDF15, GPNMB, IGFBP6) were quantified applying enzyme-linked immunosorbent assay (ELISA) techniques (DuoSet ELISA development kits, R&D systems, Wiesbaden, Germany). All measurements were performed in technical duplicates and were repeated in cases of an intra-duplicate variation higher than 20%.

### 4.4. Statistical Analysis

For explorative data analysis, a statistical software package (*SPSS 26.0*) was used. Non-parametric numerical parameters were analyzed by *Mann-Whitney U*-test (for 2 unrelated samples) or by *Wilcoxon* test (for 2 related samples). For the analysis of dynamic changes, testing with a general linear model for repeated measures was applied. Correlation analysis of parameters was performed applying non-parametric Spearman-rho test and testing for partial correlation. In order to test numerical parameters as determinants of classified variables, ROC curve analysis with AUC calculation was applied. *p* values below 0.05 (two tailed) were considered as statistically significant. In the figures, means are displayed as dots with whiskers giving the 95% CI. Box plots indicate median, upper/lower quartiles, interquartile range, minimum/maximum values, and outliers.

## 5. Conclusions

Out of five proteins with endocrine bioactivity and functions in metabolic and immune-modulatory processes, CCL5, GDF15, and IGFBP6 exhibited a significant decline associated to weight loss induced by either restrictive dietary intervention or RYGB bariatric surgery. These factors might serve as biomarkers for the therapeutical success of conventional, as well as surgical strategies, targeting morbid obesity and metabolic syndrome. In particular, GDF15 represents a valuable marker for hepatic fibrosis, arterial hypertension, and type 2 diabetes mellitus in obese individuals and, therefore, might be applicable in minor-invasive clinical diagnostics.

## Figures and Tables

**Figure 1 ijms-23-14020-f001:**
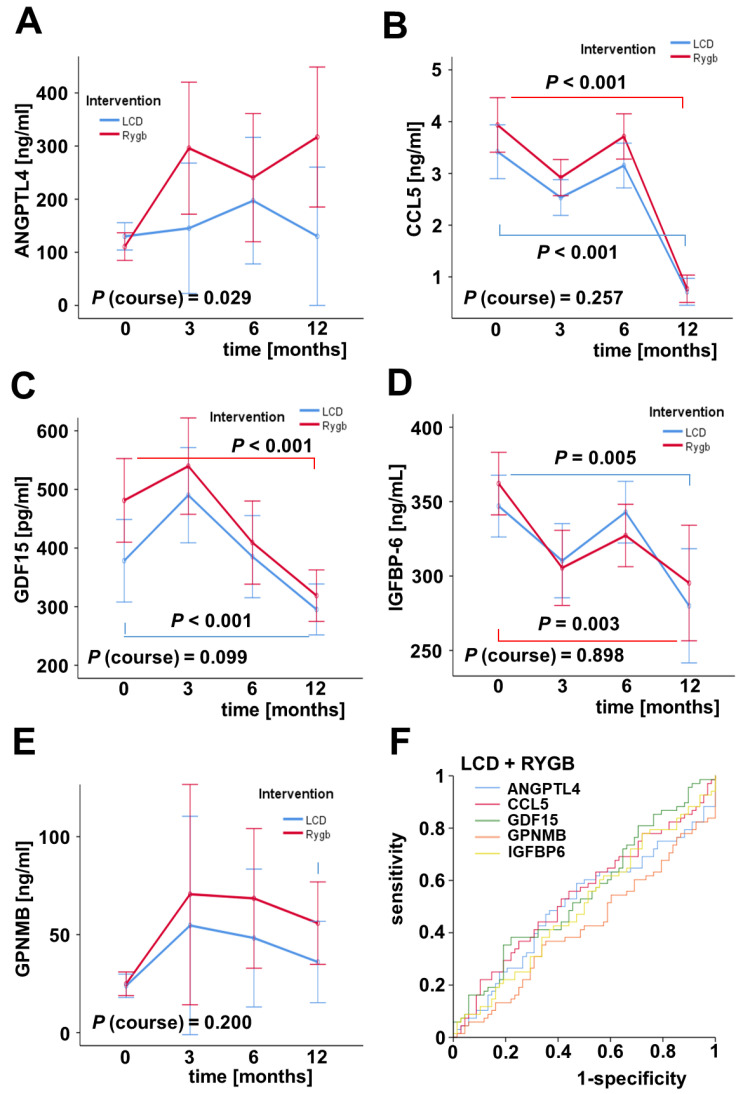
Dynamic changes in cytokine serum levels during 12 months of weight loss: ANGPTL4 (**A**), CCL5 (**B**), GDF15 (**C**), GPNMB (**D**), IGFBP6 (**E**). ROC curves of changes in ANGPTL4, CCL5, GDF15, GPNMB, and IGFBP6 levels for body fat loss classification in the whole study cohort (LCD + RYGB) (**F**).

**Figure 2 ijms-23-14020-f002:**
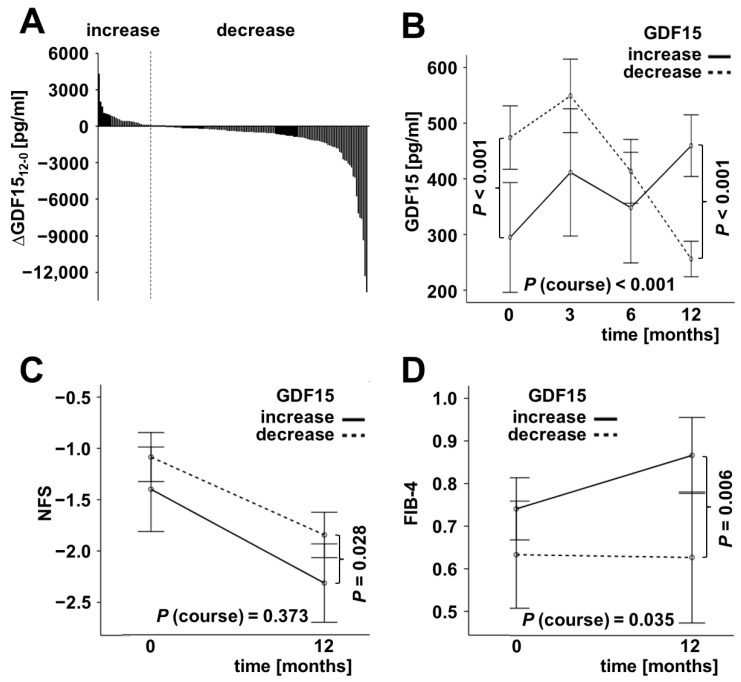
Serum GDF15 dynamics and scores of liver fibrosis. Systemic GDF15 levels increased in 40 patients and decreased in 120 patients during 12 months of weight loss (**A**) with significantly differing dynamics (**B**). Increased GDF15 concentrations were accompanied by slightly lowered NFS but higher FIB-4 values after 12 months, indicating a higher risk of fibrosis (**C**,**D**).

**Figure 3 ijms-23-14020-f003:**
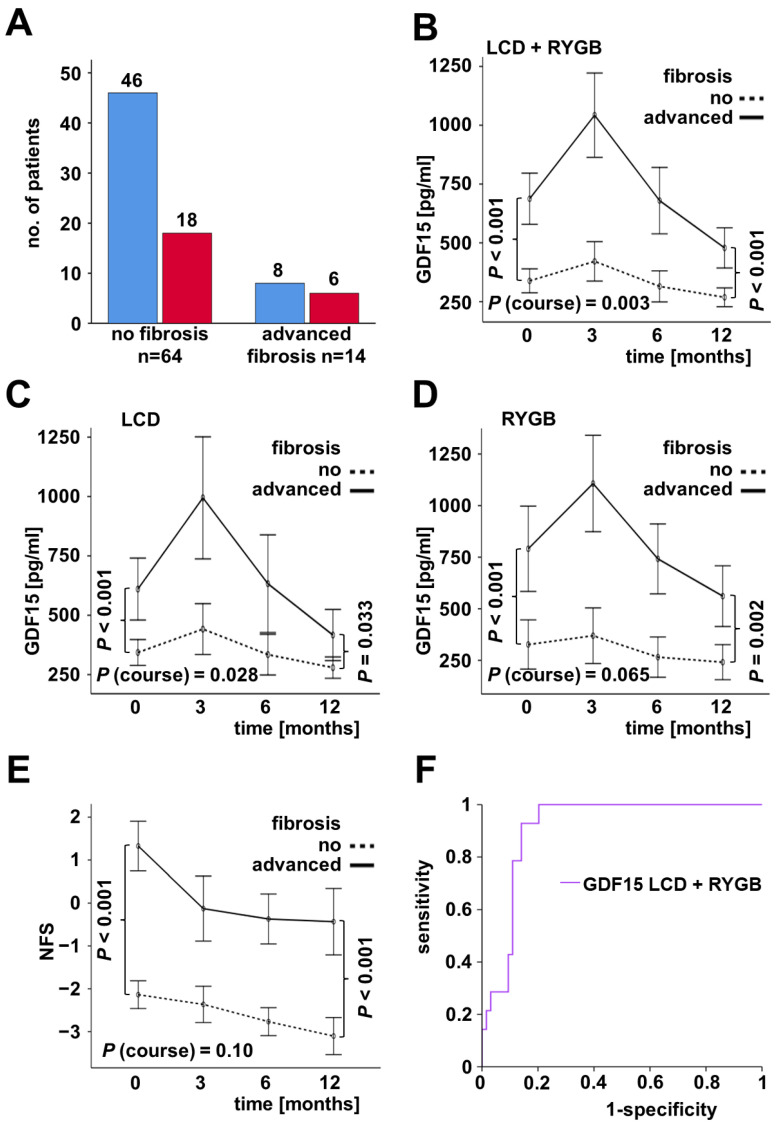
Relation between GDF15 and fibrosis. A significant proportion of patients (blue bars: LCD; red bars: RYGB) were categorized as suffering from advanced fibrosis at the study base-line (**A**). Dynamic changes in GDF15 levels during weight loss occurred exclusively in fibrotic patients (**B**) and in both study cohorts (**C**,**D**). The non-significant decrease of NAFLD fibrosis score (NFS) during the 12 months was independent of fibrosis (**E**). ROC curve of GDF15 for classification liver fibrosis (non-fibrotic vs. advanced fibrosis) in the whole study cohort (LCD + RYGB) (**F**).

**Figure 4 ijms-23-14020-f004:**
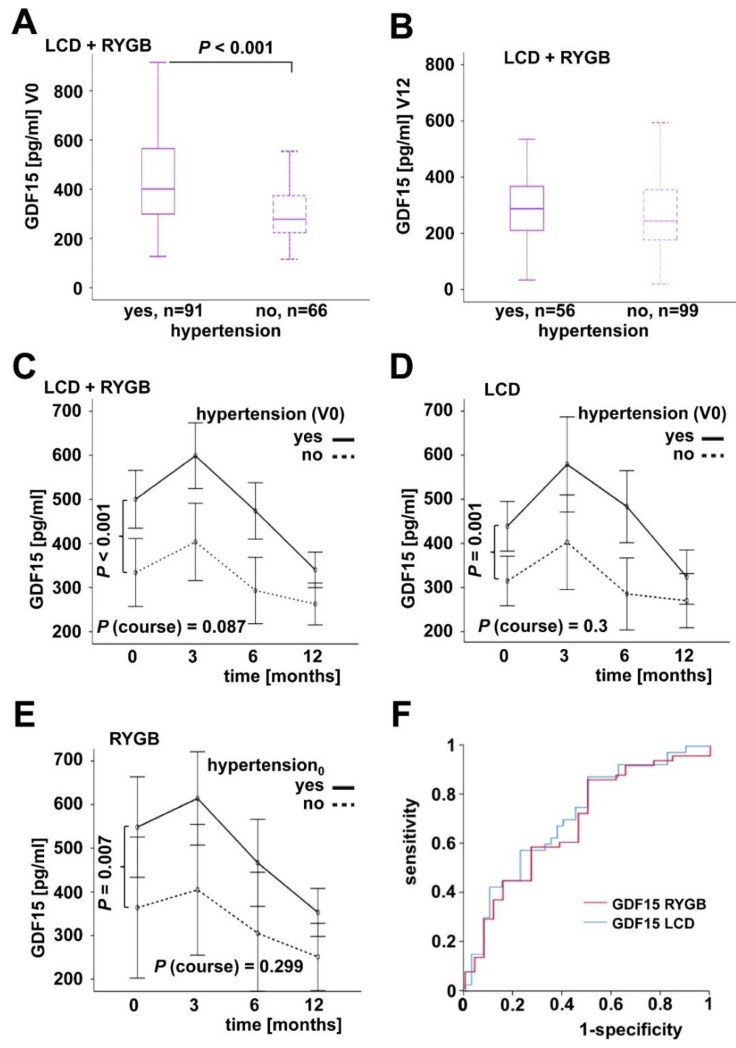
Systemic GDF15 levels in the context of hypertension. Patients with hypertension exhibited elevated base-line (V0) GDF15 concentrations (**A**), whereas no significant differences were observed at the 12 month follow-up (V12) (**B**). The decrease of GDF15 levels during the 12 months was independent of hypertension and of therapeutic strategy (**C**–**E**). Serum GDF15 is moderately predictive for classified hypertension at the study base-line (**F**).

**Figure 5 ijms-23-14020-f005:**
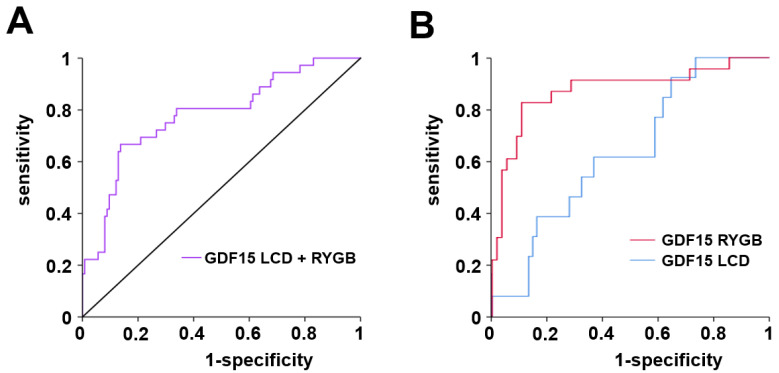
ROC analysis of base-line GDF15 concentrations with T2D. Analysis was performed for the whole study cohort (**A**) and for LCD and RYGB groups separately (**B**).

**Table 1 ijms-23-14020-t001:** Base-line and 12 month follow-up characteristics of patients in the low-calorie diet (LCD; n = 81) and the Roux-en-Y gastric bypass sub-cohort (RYGB; n = 79). Means are depicted and ranges of values are given in brackets.

Parameters	Base-Line	12 Month Follow Up	*p*
**A Low-calorie Diet**			
Age [years]	42.8 (20; 67)	-	-
Female Male	52 (64.2%) 29 (35.8%)	- -	-
BMI [kg/m**^2^**]	43.6 (31.9; 59.2)	33.5 (24.3; 49.7)	<0.001
Body weight [kg]	130 (90.1; 185.4)	99.6 (61; 159)	<0.001
Weight loss [%]	-	23 (1; 41.4)	-
Body fat [%]	45.9 (28.5; 59.2)	34.8 (15.0; 53.7)	<0.001
Waist-hip ratio	0.95 (0.69; 1.25)	0.9 (0.72; 1.13)	<0.001
**B Roux-en-Y gastric bypass**			
Age [years]	40.7 (20; 60)	-	-
Female Male	65 (82.3%) 14 (17.7%)	- -	-
BMI [kg/m**^2^**]	51.7 (42; 62)	33.1 (24; 42)	<0.001
Body weight [kg]	149.4 (109; 244)	94.6 (61; 146)	<0.001
Weight loss [%]		35.45 (16.75; 54.91)	-
Body fat [%]	52 (30; 62.1)	35.5 (19.6; 49.1)	<0.001
Waist-hip ratio	0.96 (0.71; 1.33)	0.88 (0.71; 1.05)	<0.001

## Data Availability

The data presented in this study are available on request from the corresponding author.

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
