# Peer review of "Circulating Adipokines and Hepatokines Serve as Diagnostic Markers during Obesity Therapy"

_ijms, 2022, doi:10.3390/ijms232214020_

Round 1

Reviewer 1 Report

Very well presentation for very difficult subject, congratulation for the authors

Author Response

Reply to Reviewer #1

We thank Reviewer for the kind revision of our manuscript and the positive recommendation for publication.

Reviewer 2 Report

Nice study looking at cytokine levels in patients before and after weight loss via low-calorie diet versus gastric bypass surgery.  Data demonstrating relationship of these cytokine levels to weight loss is interesting, but I do have a few concerns:

-Why were these cytokines measured and not others?  I am curious if others were explored that did not have significant results, or if these were chosen based on common mechanism?

-How were endpoints/correlates of liver fibrosis, T2D and hypertension chosen?  Was this based on hypothesized mechanism?  Were others tested and if so please add description of strategy to methods section.

-Please elaborate in discussion on possible mechanisms relating GDF15 to liver fibrosis (and other outcomes- hypertension, T2D). How do you know that GDF15 is not just a marker of obesity that we know is linked to these comorbidities?  Really, the data are most impactful if screening for liver fibrosis with GDF15 levels adds more than BMI alone- would be helpful to show this.  

Author Response

Point-to-point reply to Reviewer #2

We thank the Reviewer for the careful revision of our manuscript and for the very interesting comments that helped us to improve the quality of our manuscript significantly.

1- “Why were these cytokines measured and not others?  I am curious if others were explored that did not have significant results, or if these were chosen based on common mechanism?

Circulating GDF15 was analyzed in blood serum as a currently discussed promising factor which might serve as a therapeutic target for obesity (Day 2019) and GPNMB because of its important features in the crosstalk between liver and adipose tissue (Gong 2019). The regulation of IGFBP6, ANGPTL4, and CCL5 during weight loss was described before (Hempel et al 2021) but their predictive potential for liver fibrosis or T2D has not been addressed yet.

In order to clarify this point the above statement was introduced into the Materials and Methods section of the revised manuscript version in lines 348-352:

“Circulating GDF15 was analyzed in blood serum as a currently discussed promising factor which might serve as a therapeutic target for obesity [38] and GPNMB because of its important features in the crosstalk between liver and adipose tissue [44]. The regulation of IGFBP6, ANGPTL4, and CCL5 during weight loss was described before [45] but their predictive potential for liver fibrosis or T2D has not been addressed yet.”

2- “How were endpoints/correlates of liver fibrosis, T2D and hypertension chosen?  Was this based on hypothesized mechanism?  Were others tested and if so please add description of strategy to methods section.

12 months follow-up visits after study base-line were chosen for analysis of physiological parameters due to the observed considerable decrease in body weight and body fat percentage in both study cohorts during this time-span (Schmid 2022). We hypothesized that this general metabolic improvement might be associated with significant beneficial effects on liver fibrosis, T2D, and hypertension. Established cutoff values of NAFLD fibrosis score (NFS) were applied in order to discriminate patients with advanced (NFS > 0.675) and without liver fibrosis (NFS < -1.455) (Schmid 2022, Arnouk 2020). Patients with T2D were defined by previous medical diagnosis or by cutoff values for either HbA1c levels (≥ 6.5 %), fasting glucose (> 110 mg/dL), or impaired glucose tolerance (> 180 mg/dL glucose after 2 h in standardized oral glucose tolerance test). Individuals with arterial hypertension were identified by medical diagnosis and antihypertensive drug medication.

There was no testing of alternative endpoints and correlates.

In order to clarify the definition of endpoints in more detail, the following paragraph was added to the Materials and Methods section (lines 337-346):

“12-months follow-up visits were chosen for analysis of physiological parameters due to the observed considerable decrease in body weight and body fat percentage in both study cohorts during this time-span [30], which was hypothesized to be associated with significant beneficial effects on liver fibrosis, T2D, and hypertension. Established cutoff values of NAFLD fibrosis score (NFS) were applied in order to discriminate patients with advanced (NFS > 0.675) and without liver fibrosis (NFS < -1.455) [43]. Patients with T2D were defined by previous medical diagnosis or by cutoff values for either HbA1c levels (> 6.5 %), fasting glucose (> 110 mg/dL), or impaired glucose tolerance (> 180 mg/dL glucose after 2 h in standardized oral glucose tolerance test). Individuals with arterial hypertension were identified by medical diagnosis and antihypertensive drug medication.”

3- “Please elaborate in discussion on possible mechanisms relating GDF15 to liver fibrosis (and other outcomes- hypertension, T2D). How do you know that GDF15 is not just a marker of obesity that we know is linked to these comorbidities?  Really, the data are most impactful if screening for liver fibrosis with GDF15 levels adds more than BMI alone- would be helpful to show this.”

We thank the reviewer for this very important and helpful comment! We added an intensive discussion on GDF15 involvement in disease onset into the revised manuscript:

Discussion (lines 255-262): “Elevated GDF15 levels in hepatic fibrosis strongly argue for a significant role of this cytokine in the pathogenesis of the disease. As reported previously by Kim et al., GDF15 attenuates non-alcoholic steatohepatitis by inhibition of pro-fibrotic gene expression in mice [34]. As a regulator of appetite and nutritional behavior, elevated systemic GDF15 concentrations might further represent a mechanism for amelioration of hepatic fibrosis by restricted food intake and therefore reduced fat load in the liver. Of particular importance, data supporting a putative role of GDF15 as an obesity-independent marker and predictor of hepatic fibrosis remain scarce so far.”

Discussion (lines 292-295): “Elevated GDF15 concentrations in T2D patients might be considered as an anti-diabetic feed-back mechanism due to restricting effects on food intake and subsequent beneficial impact on weight loss [38] and insulin resistance.”

Applying partial correlation analysis with correction for BMI, we identified BMI-independent, significant and positive correlations of GDF15 with HbA1c (as a marker for T2D) and with NFS (liver fibrosis). This finding strongly argues for GDF15 serum levels as an independent biomarker for T2D as well as liver fibrosis. Subgroup analysis with respect to base-line BMI (3 groups: =< 45; 45.01 – 50; > 50) revealed significantly elevated GDF15 levels accompanying hypertension exclusively in patients with BMI above 45 kg/m2. We therefore conclude that GDF15 represents a BMI-independent biomarker for T2D and liver fibrosis whilst the association with hypertension might predominantly occur in individuals with very high BMI.

We introduced these additional findings into the manuscript’s Results and Discussion section.

Results (lines 156-157): “Partial correlation analysis revealed a positive, BMI-independent correlation of base-line GDF15 serum levels with NFS values (r = 0.392, P < 0.001).”

Results (lines 169-171): “Dividing the study cohort into subgroups defined by BMI, we found that this difference was significant for patients with BMI higher than 45 kg/m2 (n = 105) but not for individuals with lower extent of obesity (BMI ≤ 45; n = 55).”

Results (lines 190-191): “This correlation was independent of BMI in the RYGB group as was confirmed by partial correlation analysis (r = 0.556, P < 0.001) (LCD + RYGB: r = 0.432, P < 0.001).”

Discussion (lines 248-250): “Of note, base-line GDF15 serum concentrations positively correlated with NFS after correction for BMI, suggesting GDF15 as a BMI-independent marker of hepatic fibrosis in obese individuals.”

Discussion (lines 269-271): “Importantly, our present data indicate that the observed elevation of GDF15 levels in hypertension depends on BMI and might be an attribute rather of excessively obese individuals (BMI > 45 kg/m2).”

Discussion (lines 280-282): “At study base-line, GDF15 serum concentrations positively correlated with HbA1c levels – a marker for long-term elevated systemic glucose levels and insulin resistance. This correlation was independent of BMI as we confirmed by partial correlation analysis.”

Reviewer 3 Report

In this manuscript, the authors investigated the dynamics of physiologically relevant adipokines and hepatokines (ANGPTL4, 16 CCL5, GDF15, GPNMB, IGFBP6) post-intervention with low-calorie diet (LCD) and Roux-en-Y gastric bypass (RYGB), as well as their correlation with fat mass reduction and improvement of liver function. 160 severely obese patients, 81 in an LCD program whereas 79 underwent RYGB surgery, were enrolled. The authors found that systemic GDF15 was an excellent predictive marker for hepatic fibrosis as well as type 2 diabetes mellitus. The specific GDF15 cut-off values for both diseases were also calculated. The authors concluded that circulating levels of specific adipokines and hepatokines are regulated with therapy-induced fat loss and metabolic improvement and might serve as biomarkers for the success of obesity therapy strategies.

This is a large and well-charactrized obesity cohort to investigate the dynamics of physiologically relevant adipokines and hepatokines post-intervention as well as their correlation with fat mass reduction and improvement of liver function. The data were well collected and analyzed. The manuscript was well prepared. Thus, this article provides useful information for the clinicians to manage the obese patients. 

Comments

1. The authors found that systemic GDF15 was an excellent predictive marker for hepatic fibrosis and type 2 diabetes mellitus. However, the authors should discuss the roles of GDF15 in the pathogenesis of hepatic fibrosis and type 2 diabetes mellitus.

2. The authors should analyze the predictive value of adipokines and hepatokines for improvement of fat mass reduction, liver function, hepatic fibrosis and type 2 diabetes mellitus.

Author Response

Point-to-point reply to Reviewer #3

We thank the Reviewer for the careful revision of our manuscript and for the very interesting comments that helped us to improve the quality of our manuscript significantly.

1- “The authors found that systemic GDF15 was an excellent predictive marker for hepatic fibrosis and type 2 diabetes mellitus. However, the authors should discuss the roles of GDF15 in the pathogenesis of hepatic fibrosis and type 2 diabetes mellitus.

We thank the reviewer for this valuable comment and added this issue to the discussion:

Discussion (lines 255-262): “Elevated GDF15 levels in hepatic fibrosis strongly argue for a significant role of this cytokine in the pathogenesis of the disease. As reported previously by Kim et al., GDF15 attenuates non-alcoholic steatohepatitis by inhibition of pro-fibrotic gene ex-pression in mice [34]. As a regulator of appetite and nutritional behavior, elevated systemic GDF15 concentrations might further represent a mechanism for amelioration of hepatic fibrosis by restricted food intake and therefore reduced fat load in the liver. Of particular importance, data supporting a putative role of GDF15 as an obesity-independent marker and predictor of hepatic fibrosis remain scarce so far.”

Discussion (lines 292-295): “Elevated GDF15 levels in T2D patients might be considered as an anti-diabetic feed-back mechanism due to restricting effects on food intake and subsequent beneficial impact on weight loss [38] and insulin resistance.”

2- “The authors should analyze the predictive value of adipokines and hepatokines for improvement of fat mass reduction, liver function, hepatic fibrosis and type 2 diabetes mellitus.

Thank you for this important comment! We did not analyze the predictive value of adipokines and hepatokines for liver function as the initial albumin serum concentrations of all included patients were in the normal range. However, we performed correlation analysis including base-line serum concentrations of the investigated cytokines and changes in body fat percentage, NAFLD fibrosis score (NFS), and HbA1c levels in order to identify a potential predictive value for metabolic improvements. While there were no significant correlations for Angptl4, CCL5, GPNMB, and IGFBP6 with the aforementioned metabolic parameters, GDF15 was negatively correlated with changes in HbA1c levels (rho = -0.346, P < 0.001) but not with body fat loss and NFS changes. Thus, we conclude that GDF15 represents a predictor of decreasing HbA1c levels improvement of type 2 diabetes mellitus during weight loss.

We added this information to the Results and Discussion sections of the revised manuscript:

Results (lines 193-196): “Furthermore, base-line GDF15 serum concentrations were negatively correlated with changes in HbA1c levels (rho = -0.346, P < 0.001) at 12-months follow-up, unlike the other cytokines investigated in the present study.”

Discussion (lines 283-285): “Of note, basal GDF15 levels were also negatively correlated with HbA1c changes during 12 months after begin of intervention. This finding suggests circulating GDF15 as a predictor of weight loss-associated T2D amelioration.”

Round 2

Reviewer 2 Report

The authors have satisfactorily addressed concerns/questions.